# Unveiling *Acinetobacter endophylla* sp. nov.: A Specialist Endophyte from *Peganum harmala* with Distinct Genomic and Metabolic Traits

**DOI:** 10.3390/microorganisms13122843

**Published:** 2025-12-15

**Authors:** Salma Mouhib, Khadija Ait Si Mhand, Nabil Radouane, Khaoula Errafii, Issam Meftah Kadmiri, Derly Andrade-Molina, Juan Carlos Fernández-Cadena, Mohamed Hijri

**Affiliations:** 1African Genome Center, University Mohammed VI Polytechnic (UM6P), Lot 660, Hay Moulay Rachid, Ben Guerir 43150, Morocco; salma.mouhib@um6p.ma (S.M.); khadija.aitsimhand@um6p.ma (K.A.S.M.); nabil.radouane-ext@um6p.ma (N.R.); khaoula.errafii@um6p.ma (K.E.); 2Plant and Microbial Biotechnology Center, Agrobiosciences, College of Agriculture and Environmental Sciences, University Mohammed VI Polytechnic (UM6P), Lot 660, Hay Moulay Rachid, Ben Guerir 43150, Morocco; issam.kadmiri@um6p.ma; 3OMICS Sciences Laboratory, Faculty of Health Science, Universidad Espíritu Santo, Samborondón 092301, Ecuador; dmandrademolina@uees.edu.ec (D.A.-M.); 4Harvard Medical School, Brigham and Women’s Hospital, Boston, MA 02115, USA; 5Institut de Recherche en Biologie Végétale (IRBV), Département de Sciences Biologiques, Université de Montréal, 4101 Rue Sherbrooke Est., Montréal, QC H1X 2B2, Canada

**Keywords:** *Acinetobacter endophylla*, bacterial endophytes, *Peganum harmala*, plant-growth-promoting traits, secondary metabolite biosynthesis, whole genome sequencing

## Abstract

*Peganum harmala* (L.) Schrad., a perennial medicinal plant thriving in arid Moroccan soils, represents a natural reservoir of unexplored bacterial diversity. To uncover its hidden foliar endosphere microbiota, we isolated and characterized two *Acinetobacter* strains: a novel endophytic bacterium, AGC35, and another strain, AGC59, newly reported from this host. Both are non-halophilic, aerobic, Gram-negative bacteria exhibiting optimal growth at 30–35 °C, pH5, and with 1% NaCl. An integrative genomic, phylogenetic, functional, and phenotypic characterization classified both strains within the genus *Acinetobacter* (class Gamma-pseudomonadota). However, Average Nucleotide Identity (<96%) and digital DNA-DNA Hybridization (<70%) values distinguished the AGC35 strain as a novel species, for which the name *Acinetobacter endophylla* sp. nov. is proposed. A comparative genomic and phenotypic analysis with the co-isolated *Acinetobacter pittii* strain AGC59 revealed extensive genome rearrangements, reflecting distinct evolutionary lineage and ecological strategies. While both genomes share core metabolic pathways, *A. endophylla* harbors specialized genes for the degradation of plant-derived aromatic compounds (e.g., catechol) but shows reduced capacities in carbohydrate metabolism and osmotic stress tolerance, traits indicative of a metabolic specialist with plant-growth-promotion potential, including phosphorus, potassium, and silicon solubilization and indole-3-acetic acid production. In contrast, *A. pittii* exhibits a more generalist genome enriched in stress functions. Analysis using the Antibiotics and Secondary Metabolite Analysis Shell revealed multiple biosynthetic gene clusters in both strains, showing ≤26% similarity to known references, suggesting the potential for novel antimicrobial secondary metabolite biosynthesis, including antifungal lipopeptides and thiopeptide antibiotics. Altogether, functional specialization and ecological coherence of these findings support the recognition of *A. endophylla* sp. nov. as a genomically and functionally distinct species, highlighting niche partitioning and adaptive metabolism within the *P. harmala* holobiont. These results underscore the plant’s value as a reservoir of untapped microbial diversity with significant ecological and biotechnological relevance. Finally, future work will focus on elucidating the novel metabolites encoded by the biosynthetic gene clusters in both isolates and exploring their applications in crop-improvement strategies and natural-product discovery.

## 1. Introduction

The plant holobiont, defined as an integrated ecological and functional unit composed of plant host and its associated microbiota, represents a central paradigm in modern microbial ecology [1]. Among these microbial partners, endophytes, microorganisms that inhabit the internal tissues of healthy plants without causing disease, play pivotal roles in nutrient cycling, stress tolerance, and host health regulation through the production of bioactive metabolites [2]. The ecological and functional dynamics of bacterial endophytes are increasingly recognized as key drivers of plant resilience under abiotic stress, particularly in non-model plants inhabiting extreme environments [3].

*Peganum harmala* L. (Zygophyllaceae) is a perennial xerophytic medicinal plant native to arid and semi-arid regions. It thrives in marginal, saline, alkaline, and drought-prone soils, reflecting remarkable ecological plasticity. The species is also known for producing a wide array of β-carboline and quinazoline alkaloids (e.g., harmine, harmaline), which may influence its symbiotic associations within the holobiont [4,5]. Despite extensive pharmacological studies, the endospheric microbiota of *P. harmala* remains largely unexplored. Its chemically rich and stress-adapted habitat represents an ideal niche for the discovery of novel microorganisms with adaptive traits and biotechnological potential, shedding light on microbial evolution and plant–microbe interactions in hyper-arid ecosystems.

Among potential endophytes, members of the genus *Acinetobacter* (family Moraxellaceae) exhibit remarkable ecological diversity [6], ranging from opportunistic hospital pathogens, such as *A. baumannii*, to beneficial environmental and plant-associated taxa including *A. johnsonii, A. pittii, A. calcoaceticus,* and *A. junii* [7,8,9,10]. These bacteria are metabolically versatile, capable of surviving under various environmental stresses and utilizing diverse substrates. However, most environmental and plant-associated *Acinetobacter* strains remain poorly studied, and the genomic foundations of their adaptation to the plant endosphere are still largely unknown [11].

The discovery and characterization of novel bacterial species from underexplored ecological niches such as medicinal plants are essential for expanding our understanding of microbial diversity and uncovering new genetic resources. Medicinal plants, particularly those thriving in arid and semi-arid ecosystems, represent unique ecological reservoirs where microorganisms have evolved adaptive traits to withstand multiple environmental stresses, including drought, salinity, and nutrient scarcity. Studies consistently reveal that medicinal plants in these harsh regions host highly diverse bacterial communities, including previously uncharacterized taxa. Dominant phyla include *Actinobacteria*, *Proteobacteria*, *Bacteroidetes*, *Firmicutes*, *Acidobacteria*, and *Gemmatimonadetes*, with many isolates exhibiting distinct genetic and metabolic profiles [12,13,14,15,16,17]. Novel species, particularly within *Actinobacteria* and *Betaproteobacteria*, are frequently detected but remain underexplored for their functional potential [13,17,18]. Recent advances in genomics and polyphasic taxonomy have further highlighted that extreme habitats can harbor specialized endophytic communities with promising applications in agriculture, biotechnology, and biomedicine. For instance, Ait Si Mhand et al. [19] reported the isolation and genomic characterization of two novel *Microbacterium* species, *Microbacterium xerophyticum* sp. nov. and *Microbacterium umsixpiens* sp. nov., from the medicinal desert plant *Citrullus colocynthis* in central Morocco. Their study demonstrated that these bacteria possess genomic features associated with oxidative and osmotic stress tolerance, metal homeostasis, and phosphate acquisition, emphasizing the potential of desert plant endophytes as sources of adaptive genes shaped by extreme environmental pressures. Building on this framework, we hypothesized that the foliar endosphere of *Peganum harmala*, another medicinal plant well adapted to Moroccan drylands, harbors unique bacterial species with genomic and functional traits reflective of their host’s extreme environment [19].

In this study, we conducted a comparative genomic and functional analysis of two endophytic *Acinetobacter* isolates obtained from the leaves of *P. harmala*: a novel strain, *Acinetobacter endophylla* AGC35, and a co-isolated relative, *A. pittii* AGC59. Through an integrative approach combining whole-genome sequencing, phylogenomics, synteny analysis, KEGG functional profiling, secondary metabolite prediction, plant-growth-promoting (PGP) trait assessment, and phenotypic characterization, we aimed to define their taxonomic status and decipher their ecological strategies.

Our findings uncover *Acinetobacter endophylla* sp. nov. AGC35 as a new species and report *A. pittii* AGC59 for the first time from *P. harmala*. Together, they reveal the hidden diversity within the *P. harmala* holobiont, highlighting the evolutionary flexibility and adaptive metabolism of *Acinetobacter* species in colonizing plant niches, and providing new insights into bacterial adaptation in arid ecosystems.

## 2. Materials and Methods

### 2.1. Plant Habitat and Isolation of Endophytic Bacteria

Two *Acinetobacter* strains were isolated from the surface-sterilized, healthy vegetative shoots of *P. harmala* collected at the flowering stage in Nzalat Laadam, Benguerir, Morocco (32°06′49.6″ N 7°57′13.0″ W), on 27 May 2022 (Figure 1a), where natural populations grow spontaneously. Complete *P. harmala* plants were sampled along a 1 km transect (Figure 1b). A total of 52 healthy specimens were placed in a plastic Ziplock bag and transported in a cooler with icepacks to the laboratory (Figure 1c).

Upon arrival, plant debris was removed by rinsing with tap water. Shoots were aseptically separated with a flame-sterilized scalpel, air-dried and stored at 4 °C for immediate endophyte isolation (Figure 2a). Leaf tissues were surface-sterilized using a three-step procedure: (i) immersion in 2% (*v*/*v*) Tween 20 (Sigma-Aldrich, St. Louis, MO, USA) for 3 min, followed by 1 min rinse in sterile distilled water (SDW); (ii) immersion in 2% sodium hypochlorite for 3 min; (iii) three successive washes with SDW. The sterilized tissues were dried on sterile paper towels (Figure 2b,c).

Sterilized surface leaf fragments (~5 mm^2^) were aseptically cut using a sterile scalpel (Figure 2b) [20,21,22,23]. The efficiency of surface sterilization was verified by imprinting sterilized tissue onto agar plates and plating the final rinse water (Figure 2c). The absence of microbial growth confirmed successful disinfection [20,21,22,23,24]. Sterilized tissues were plated onto Tryptic Soy Agar (TSA) medium (per liter: pancreatic digest of casein 17 g, papaic digest of soybean meal 3 g, NaCl 5 g, K_2_HPO_4_ 2.5 g, glucose 2.5 g, and agar 15 g) and Potato Dextrose Agar (PDA) (per liter: potato extract 4 g, glucose 20 g, bacteriological agar 15 g) (Oxoid, Thermo-Fisher, Temara, Morocco), at full strength (1×) and one-tenth dilution (1/10×), using the tissue-plating method (Figure 2c). Each Petri dish was divided into four quadrants: one served as an imprint control, while the remaining three were used to place sterilized tissue fragments, each representing one replicate. In one of these quadrants, fragments were overlaid with sugar-free minimal (M) broth medium [25]. Plates were incubated at 28 °C for one month (Figure 2c). Emerging colonies with distinct morphologies were periodically sub-cultured until pure. Purified isolates were stored in 20% (*v*/*v*) glycerol in Tryptic Soy Broth (TSB) at −80 °C for long-term preservation [24].

### 2.2. DNA Extraction and Sequencing

Genomic DNA was extracted from fresh bacterial liquid cultures following Llop’s protocol [26]. Molecular identification was conducted through PCR amplification and Sanger sequencing of the 16S rRNA gene using the universal bacterial primers pAF (5′-AGA GTT TGA TCC TGG CTC AG-3′) and 926R (5′-CCG YCA ATT YMT TTR AGT TT-3′) (Eurogentec, Seraing, Belgium) [27,28]. Sequencing was performed by a commercial service at the Centre d’Expertise et de Service (Genome Quebec, Montreal, QC, Canada). PCR reactions were carried out in a Mastercycler X50s (Eppendorf, Hamburg, Germany), with an initial denaturation at 95 °C for 5 min, followed by 30 cycles of denaturation at 94 °C for 30 s, annealing at 58 °C for 30 s, and extension at 72 °C for 1 min. Each reaction had a total volume of 50 µL and was prepared using Taq DNA Polymerase (Qiagen, Global Diagnostic Distribution, Témara, Morocco) according to the manufacturer’s instructions, containing 2.5 µL 10× reaction buffer, 2.5 µL MgCl_2_ (25 mM), 2.5 µL forward primer (5 µM), 2.5 µL reverse primer (5 µM), 0.2 µL *Taq* polymerase, 37.5 µL nuclease-free water, and 1.5 µL genomic DNA template from each isolate. Amplicons were assembled using Geneious Prime v2023.0.3 (Biomatters, Auckland, New Zealand) and compared for sequence homology using the BLAST nucleotide search in the NCBI database. Two strains showing distinct colony morphologies and <98% sequence identity to known taxa were selected for whole-genome sequencing (WGS). Genomic libraries were prepared using the Nextera XT kit (Illumina, MegaFlex, Casablanca, Morocco) with 200 ng DNA input per sample and unique dual indices. Libraries were sequenced individually in a 2 × 151 bp paired-end format on an Illumina NextSeq 550 instrument, achieving a cluster density of 202 K/mm^2^. Demultiplexing was performed using bcl2fastq software (v2.17.1.14; Illumina, San Diego, CA, USA), ensuring accurate assignment for downstream analysis.

### 2.3. Bioinformatics Analysis

Figure 3 summarizes our complete workflow focusing on the bioinformatics pipeline from raw sequence data to species delineation. Low-quality reads (Q ≤ 20, length ≤ 100 bp) were removed using BBDuk74 [29]. Taxonomic assignment and contamination screening were performed using Kaiju v1.10.1 [30]. *De novo* genome assembly was performed with MaSuRCA v4.1.4 [31]. Assembly quality was evaluated with QUAST v5.3.0 [32], selecting the draft with fewest contigs and highest N50 value. Genome completeness and contamination were assessed using CheckM v1.2.0 [33]. If contamination or fragmentation was detected, reassembly was performed with metaSPAdes v4.2.0 [34]. Prior to assembly, read binning was conducted with MaxBin2 v2.2.7 [35] and DAS Tool v1.1.7 [36]. Final assemblies were validated with a second round of QUAST v5.3.0 and CheckM analyses.

### 2.4. Taxonomic Assignment and Phylogenomic Analysis

Assembled genomes were analyzed using Type (Strain) Genome Server (TYGS) [37] and PubMLST https://pubmlst.org [38] (accessed on 11 December 2025). TYGS compared whole-genome and 16S rRNA sequences against type strains to identify related taxa, while PubMLST assigned isolates based on 53 ribosomal protein subunits (*rps* genes) using a 95% similarity threshold.

Average Nucleotide Identity (ANI) was computed using FastANI v1.33, applying a 96% species threshold [39]. Digital DNA–DNA hybridization (dDDH) was estimated using the Genome-to-Genome Distance Calculator with a 70% cutoff for species delineation [40].

A phylogenomic tree was constructed from 36 complete *Acinetobacter* genomes, including AGC35, AGC59, and closely related type strains, using Average Nucleotide Identity (ANI). Pairwise ANI values were calculated with FastANI v1.33 and converted into evolutionary distances using the following formula: Distance = 1 − (ANI/100). The resulting symmetric distance matrix was used as input for FastME v2.1.6.3 to reconstruct a minimum-evolution tree under the neighbor-joining framework. Since ANI values represent whole-genome similarity rather than position-specific alignments, bootstrap resampling cannot be applied, and therefore, no bootstrap values are included. The final tree was rooted using the midpoint method and annotated to include relevant *Acinetobacter* type strains before visualization with FastME v2.1.6.3 under the Neighbor-Joining (Minimum Evolution) algorithm. *Psychrobacter arcticus* 273-4 (NC_007204) served as the outgroup. The consensus tree was visualized and manually re-rooted in Geneious Prime v 2023.0.3 (Biomatters, Auckland, New Zealand).

### 2.5. Functional Annotation, Genome Visualization, and Comparative Genomics

Genome annotation was performed using Prokka [41]. Functional assignment was based on KEGG Orthologs (KOs) via KofamKOALA [42] and RAST Subsystems. Comparative genomic analyses were performed using the RAST https://rast.nmpdr.org (accessed on 11 December 2025) subsystems database [43], and then core metabolic pathways and gene counts were grouped into major functional categories and visualized as Sankey diagrams and heatmaps.

Endophytic trait-associated genes were curated from Bhardwaj et al. [44] and Morobane et al. [45], including genes linked to amino acid and carbohydrate metabolism, macro-element utilization, energy production, cellular defense, and virulence. Gene counts were normalized using Z-scores and visualized in RAWGraphs 2.0 as a blue–orange heatmap.

Secondary metabolite biosynthetic gene clusters (BGGs) were predicted using antiSMASH [46] with default parameters for NRPS/PKS and RiPP detection. The draft genome of the novel *Acinetobacter* AGC35 genome was visualized using Proksee https://proksee.ca (accessed on 11 December 2025) [47] as a circular genome map highlighting GC content, GC skew, and genomic islands enriched in mobile elements and antimicrobial resistance genes. Annotation cross-references were made with MobileOG-db [48] and CARD [49].

Pairwise whole-genome alignments were generated with Mauve (progressiveMauve module; default scoring function, “Move contigs”). The larger, high-weight locally collinear blocks (LCBs) were selected to evaluate synteny conservation and assembly quality. Draft assemblies (AGC35 and AGC59) in FASTA format were aligned with their closest related species, and the resulting LCBs were visualized with the Mauve viewer and exported as PNG images.

### 2.6. Phenotypic Profiling

Phenotypic profiling was conducted using the Biolog GEN III system (Biolog Inc., Hayward, CA, USA), following manufacturer instructions. The system tests the utilization of 71 carbon sources and resistance to 23 chemicals via tetrazolium-based colorimetric reactions. Plates were incubated at 28 °C for 72 h, and absorbance was measured at 590 nm using the Biolog MicroStation™ (Biolog, Inc., Hayward, CA, USA). Results were interpreted using the GEN III species database [50].

### 2.7. Plant-Growth-Promoting (PGP) Characterization

Each isolate was cultured in TSB for 24 h at 28° C under agitation (150 rpm), harvested by centrifugation (8000 rpm, 5 min), washed with PBS, and adjusted to OD_600_ = 0.8. Phosphate solubilization was assessed using the National Botanical Institute’s phosphate (NBRIP) growth medium [51]. Potassium solubilization was tested on Aleksandrov medium containing mica as the potassium source [52]. Zinc and silicate solubilization were evaluated on nutrient agar supplemented with ZnO and silicate, respectively, using bromothymol blue or bromocresol purple as pH indicators [53,54]

Plates were incubated at 28 °C for 5–7 days; color change and halo formation indicated mineral solubilization.

Indole-3-acetic acid (IAA) production by the bacterial isolates was assessed using a qualitative assay, following the method by Patten and Glick [55]. Cultures were grown in DF minimal medium supplemented with 1 mg mL^−1^ tryptophan for 48 h at 28 °C and then mixed with Salkowski reagent; the development of a red–pink coloration indicated IAA synthesis. Nitrogen utilization was evaluated by streaking isolates onto nitrogen-deficient combined carbon (CC) medium [56], with visible growth after 7 days of incubation at 28 °C considered positive for nitrogen assimilation. All PGP assays were performed with four replicates for each bacterial strain.

## 3. Results

### 3.1. Isolation of Bacterial Endophytes

The isolation procedure yielded two *Acinetobacter* strains from the foliar tissues of *Peganum harmala* (Ph-F): AGC35 (Figure 4a) and AGC59 (Figure 4b). The strains originated from different plant samples; AGC35 was isolated on TSA, whereas AGC59 unexpectedly appeared on PDA, a medium initially intended for fungal endophytes. Both isolates exhibited clearly distinct colony morphologies.

### 3.2. Genome Insights of Acinetobacter Strains

#### 3.2.1. Phylogenomics and Genome Structure

Post-taxonomic classification of *Acinetobacter* isolates from *P. harmala* (Table 1), aphylogenomic analysis of 36 *Acinetobacter* genomes placed AGC59 firmly within the *A. pittii* clade, clustering near strain PHEA-2, whereas AGC35 formed a distinct monophyletic lineage sister to, but outside, the *A. lwoffii* group (Figure 5). These relationships were supported by ProgressiveMauve alignments (Figure 6). AGC35 showed extensive genomic rearrangements relative to *A. lwoffii* H7, comprising 58 locally collinear blocks that reflect multiple inversions and translocations. In contrast, AGC59 displayed a largely conserved syntenic architecture with *A. pittii* PHEA-2, with only 15 LCBs. The magnitude of structural divergence in AGC35 is consistent with species-level separation rather than variation within *A. lwoffii* H7. Thus, the main endophytic traits of both strains AGC35 and AGC59 are illustrated in Appendix A.

#### 3.2.2. Genome Assembly, Quality, and Taxonomic Assignment

The draft genome of AGC35 (*Acinetobacter endophylla* sp. nov.) is 3.75 Mb (329 contigs; N50 = 72,460 bp; GC = 42.58%) and showed 99.93% completeness with 0.72% contamination at 73× coverage (Appendix A). The 16S rRNA gene exhibited 97.8% similarity to its closest validly named *Acinetobacter* type strain (PV739381). Genomic comparisons yielded 96% ANI and 64.5% dDDH relative to *A. lwoffii,* similar to 16S rRNA gene Sanger sequencing outputs (Appendix A), falling below species-delimitation thresholds, for which the name *Acinetobacter endophylla* sp. nov. is proposed. MLST returned no allele match, and TYGS did not affiliate AGC35 with any recognized type strain, indicating a novel genomic lineage. These results collectively support its designation as a new species. Genome data are deposited under SRA accession SRR29855794. Whole-genome alignments using ProgressiveMauve show a significant rearrangement, including inversion and translocation in AGC35 relative to *A. lwoffii* H7 (58 locally collinear blocks), whereas AGC59 displays a conserved syntenic backbone with *A. pittii* PHEA-2 (15 LCBs) (Figure 6). The degree and pattern of genome structural reallocation in AGC35 are consistent with a distinct genomic species rather than a simple strain variant.

### 3.3. Functional and Metabolic Potential Abilities

KOfamKOALA and KEGG annotations showed that both genomes encode the typical core functions of *Acinetobacter*, including respiration, central carbon metabolism, biosynthesis of nucleotides and amino acids, and general stress responses (Figure 7 and Appendix A, Appendix A). Despite these shared features, each strain displayed clear functional enrichments. *A. pittii* AGC59 exhibited higher numbers of genes associated with membrane transport systems and carbohydrate metabolism, particularly glycolate/glyoxylate interconversions, trehalose metabolism, and glycerate utilization. In contrast, *A. endophylla* AGC35 was enriched in amino acid and protein metabolism, as well as cofactor and vitamin biosynthetic pathways. Across both genomes, subsystem annotations were dominated by contributions to carbohydrate (~540), amino acid (~580), energy (~450), and cofactor/vitamin (~540) metabolism (Figure 7b).

Comparative profiles based on the SEED database further emphasized their contrasting ecological strategies. AGC35 showed specialization in aromatic compound degradation, notably the catechol branch of the β-ketoadipate pathway (Z = +1.00), while exhibiting reduced capacity for central carbohydrate metabolism, such as D-gluconate utilization (Z = −2.32) and pyruvate–alanine–serine interconversions (Z = −2.68). In contrast, AGC59 displayed the characteristics of a metabolic generalist, with broader carbohydrate utilization, greater osmotic stress tolerance, and higher potential for hydroxybenzoate degradation (Z = +1.50) (Figure 7b).

antiSMASH analysis revealed diverse biosynthetic gene clusters (Figure 8) in both genomes. Each strain possessed clusters related to antifungal lipopeptides (e.g., fengycin and plipastatin) and thiopeptide antibiotics (e.g., berninamycin), although with low gene-level similarity to reference clusters, suggesting mosaic or divergent architectures. AGC35 uniquely harbored NRPS–PKS hybrid clusters and iturin/bacillomycin-like loci with 13–22% similarity to known BGCs, while AGC59 contained a berninamycin-like region with approximately 26% similarity. These low-similarity signatures indicate notable biosynthetic novelty across both strains.

### 3.4. Nomenclature and Biochemical Profiling of Novel Acinetobacter

#### *Acinetobacter endophylla* sp. nov.

*Acinetobacter endophylla* AGC35 was isolated from the leaves of *Peganum harmala*. Growth occurred on TSA at 28–35 °C within 48 h. Colonies were small (~1 mm in diameter), circular, raised, smooth, slightly translucent, and creamy-white with entire margins (Figure 4a,b). Cells observed under light microscopy were Gram-negative and non-motile coccobacilli, and no spores were detected. All phenotypic assays were performed at ~28–30 °C under aerobic conditions.

Etymology: *endophylla* (fem. adj.) from the Greek endon (“inside”) and phyton (“plant”), referring to its endophytic lifestyle within plant tissues.Species name: *Acinetobacter endophylla* sp. nov.Type strain: AGC 35^T^ (=CCMM B1335^T^).Accession numbers: 16S rRNA gene, PV739381; whole genome, SRR29855794.BioProject: PRJNA1133887; Biosample: SRP520430.DNA G + C content: 42.58%; Genome size: 3.75 Mb.ANI/closest relative: 96%. dDDH: 64.5%.

### 3.5. Biochemical, and Plant-Growth-Promoting (PGP) Characterization

As AGC35 was confirmed as a novel species, its biochemical phenotype was characterized using the Biolog GEN III MicroPlate (Figure 9). The isolate exhibited an oxidative, non-fermentative metabolic profile typical of *Acinetobacter*, with no catabolism of most mono- and disaccharides, polyols, sugar derivatives, or sugar-phosphates (Appendix A). In contrast, AGC35 showed positive oxidative respiration with inosine, L-alanine, methyl-pyruvate (G2), L-lactate (G4), Tween-40 (H1), γ-aminobutyric acid (H2), α-hydroxybutyrate (H3), β-hydroxy-DL-butyrate (H4), α-ketobutyrate (H5), propionic acid (H7), and acetic acid (H8), and it grew in 1% sodium lactate (C10). This profile indicates a metabolic preference for amino acids, short-chain organic acids, and lipid/ester substrates over carbohydrates (Appendix A).

In the stress-chemical panel, the isolate grew at pH 6 and pH 5 (A11–A12) and tolerated 1% NaCl (B10), but not higher salinities (4–8%, B11–B12). It exhibited tolerance to rifamycin SV (D11), tetrazolium blue (F12), and potassium tellurite (G12), with borderline tolerance to lincomycin (E10), Niaproof 4 (E12), and aztreonam (H10). Inhibition was observed with troleandomycin (D10), fusidic acid (C11), minocycline (D12), vancomycin (F10), nalidixic acid (G10), lithium chloride (G11), tetrazolium violet (F11), sodium butyrate (H11), and sodium bromate (H12) (Appendix A).

Using the GEN III identification database, AGC35 was placed within the genus *Acinetobacter*, closest to *A. schindleri* (PROB 0.855; SIM 0.834; DIST 2.357), but it diverged in multiple substrate and chemical responses, supporting a distinct metabolic fingerprint consistent with its status as a new species.

Qualitative functional assays further demonstrated its role as a plant-beneficial endophyte. *A. endophylla* AGC35 exhibited several PGP traits, including nitrogen fixation, solubilization of insoluble potassium (halo formation), silicate solubilization, growth on inorganic phosphate and zinc media, and auxin production (Figure 10). In contrast, AGC59 did not display substantial PGP potential. These phenotypic results are consistent with KEGG functional annotations indicating macro-element solubilization and multiple plant-associated metabolic capacities (Figure 7b).

## 4. Discussion

Genomic insights and niche specialization emerge clearly from our integrated genomic and phenotypic analyses, which position *Acinetobacter endophylla* sp. nov. AGC35 as a clear example of ecological specialization within the *Acinetobacter* genus [57]. Phylogenomic placement, combined with a highly rearranged genome organization relative to *A. lwoffii* H7 and ANI/dDDH values well below species thresholds, clearly supports its designation as a novel species. Comparative genomics with its co-isolated relative, *A. pittii* AGC59, an environmental species within the *A. calcoaceticus–baumannii* complex known for its metabolic versatility, biodegradation capacity, and multidrug resistance [58,59], provides rare genomic insight into niche partitioning within the foliar endosphere microbiota of *Peganum harmala*. These findings reveal contrasting evolutionary strategies between a specialist endophyte (*A. endophylla*) and a generalist colonizer (*A. pittii*).

Both genomes share core features typical of the genus, including transporters, stress response mechanisms, cofactor/vitamin biosynthesis, and energy metabolism, reflecting the metabolic diversity and environmental resilience of *Acinetobacter*. Notably, *A. endophylla* displays patterns of reductive evolution and functional specialization, with enrichment in amino acid and protein metabolism and depletion of pathways for diverse carbon sources (e.g., D-gluconate). Its specialization in the catechol branch of the β-ketoadipate pathway suggests adaptation to host-derived phenolic compounds, reflecting a “metabolic pruning” strategy within the nutrient-poor, carbon-limited intercellular environment of the leaf. In contrast, *A. pittii* exhibits genomic enrichment for carbohydrate metabolism and transport functions, consistent with a copiotrophic, generalist “jack-of-all-trades” lifestyle (Figure 7).

In parallel, biosynthetic potential and antagonistic traits of both strains are strongly reflected in their genomic architecture, as each harbors biosynthetic gene clusters (BGCs) that potentially encode novel bioactive metabolites. These include antifungal lipopeptides (e.g., fengycin, plipastatin, iturin/bacillomycin) and antibacterial thiopeptides (e.g., berninamycin A), with low similarity to known clusters, suggesting uncharacterized or mosaic architectures. Such compounds likely facilitate niche establishment by suppressing competitors, supporting ecological roles for microbial antagonism in the *P. harmala* foliar microbiota [10,60,61].

From genome to phenotype, functional validation consistently supports the ecological specialization of *A. endophylla*. Phenotypic profiling reveals a narrow carbon-utilization spectrum on standardized biochemical panels, indicating limited capacity to metabolize diverse or complex substrates. This restricted metabolic breadth aligns with adaptation to the endo-foliar niche, where bacteria predominantly encounter simple host-derived compounds such as monosaccharides, organic acids, and amino acids rather than heterogeneous environmental carbon sources. In addition to its limited carbon utilization, *A. endophylla* displays sensitivity to osmotic stress and tolerance to specific ionic stresses (sodium butyrate, sodium bromate, Niaproof 4), further reflecting selective pressures characteristic of the foliar environment, where microbes must rely on host-derived nutrients and withstand biotic stresses, as reported by Mujumdar et al. [62]. Its plant-growth-promoting traits, including nitrogen assimilation, solubilization of K, P, Zn, and silicate, and IAA production, support its potential as a candidate for enhancing plant growth and nutrient acquisition, similar to closely related species within the same genus [7,9]. Genomic analyses reinforce these functional inferences, revealing phosphate and polyphosphate metabolism modules, metal ion transport and efflux systems, redox enzymes, and amino acid/tryptophan biosynthetic clusters in both genomes (Figure 7b). In contrast, *A. pittii* exhibits broader metabolic versatility, consistent with a generalist ecological strategy. Together, these results underscore the value of a polyphasic framework for linking genomic content with functional capabilities [11].

In this context, the ecological and environmental significance of *Acinetobacter* becomes increasingly evident. The *Acinetobacter* genus is ecologically versatile, occurring in soils, freshwater, marine sediments, wastewater, and floral nectar and demonstrating resilience to desiccation and environmental stresses [63,64,65,66]. Many species contribute to the biodegradation of hydrocarbons, aromatic compounds, phenol, biphenyl, and other xenobiotics, supporting bioremediation and ecosystem restoration [67,68,69,70]. They also participate in nutrient cycling, including nitrogen and sulfur turnover, enhancing ecosystem function and resilience [65,66,71]. The genus displays remarkable genetic and metabolic diversity, facilitating adaptation to novel ecological niches, with specialized functions such as pectin degradation in nectar that may influence plant–microbe–pollinator interactions [72,73]. While *Acinetobacter* is occasionally found in plant microbiomes, its functional impact in these systems remains underexplored, highlighting the potential of endophytes like *A. endophylla* as reservoirs of novel bioactive compounds and plant-beneficial traits.

Altogether, implications for plant–microbe interactions and biotechnological potential become evident. The co-occurrence of specialized metabolic pathways, biosynthetic potential, and plant-growth-promoting traits *in A. endophylla* AGC35 illustrates the multifunctional role of endophytes in enhancing host resilience to biotic and abiotic stress. By contrast, *A. pittii* reflects a generalist strategy with broad metabolic and stress response capacities. Together, these findings reveal distinct ecological strategies, emphasizing the value of medicinal plants such as *P. harmala* as niches for discovering microbial diversity with potential applications in agriculture, bioremediation, and drug discovery.

Finally, the sustainable agricultural potential of *A. endophylla* sp. nov. and *A. pittii* warrants deeper investigation using multi-omics approaches (transcriptomics, metabolomics, and proteomics) to validate the expression of key plant-growth-promoting traits under plant-associated conditions. In addition, chemical profiling and structural characterization of the products encoded by their novel biosynthetic gene clusters will be essential for identifying previously unknown bioactive metabolites, including potential lipopeptides or RiPP-derived compounds. These efforts, combined with greenhouse and field-scale evaluations, will clarify their roles in biocontrol, nutrient mobilization, and stress mitigation, ultimately positioning these endophytes as promising candidates for next-generation microbial inoculants in sustainable agriculture.

## 5. Conclusions

This study reveals the foliar endophytic bacteria of *P. harmala* as a reservoir of diverse and ecologically adapted *Acinetobacter* lineages. Comparative phylogenomics clearly establishes *Acinetobacter endophylla* sp. nov. AGC35 as a new species, markedly divergent from *A. lwoffii* based on multi-metric thresholds (ANI/dDDH), while AGC59 is confirmed as *A. pittii*. Beyond taxonomic resolution, our integrated genomic and phenotypic analyses uncover distinct ecological strategies: *A. pittii* AGC59 exhibits a generalist, metabolically versatile profile, whereas *A. endophylla* AGC35 displays a specialized metabolic configuration centered on host-derived aromatic compounds, alongside robust plant-growth-promoting traits, including multi-mineral solubilization and auxin production. A compatibility test performed on TSA medium with both strains showed no growth inhibition at their intersection, indicating that the two isolates are compatible (Appendix A).

Both strains harbor rich and potentially novel biosynthetic gene clusters for antifungal and antibacterial metabolites, underscoring the ecological significance of endophyte–host interactions within the *P. harmala* holobiont and highlighting this plant as a promising source of uncharacterized natural products. Multi-omics approaches will be essential to validate the novelty and bioactivity of these compounds.

Overall, this work expands the ecological and functional landscape of the genus *Acinetobacter* and positions medicinal plants, particularly *P. harmala*, as valuable niches for discovering metabolically specialized, biotechnologically relevant endophytes. The complementary ecological identities of *A. endophylla* and *A. pittii* exemplify the hidden microbial diversity and functional potential residing within a single leaf, offering promising avenues for applications in agriculture and drug discovery.

## Figures and Tables

**Figure 1 microorganisms-13-02843-f001:**
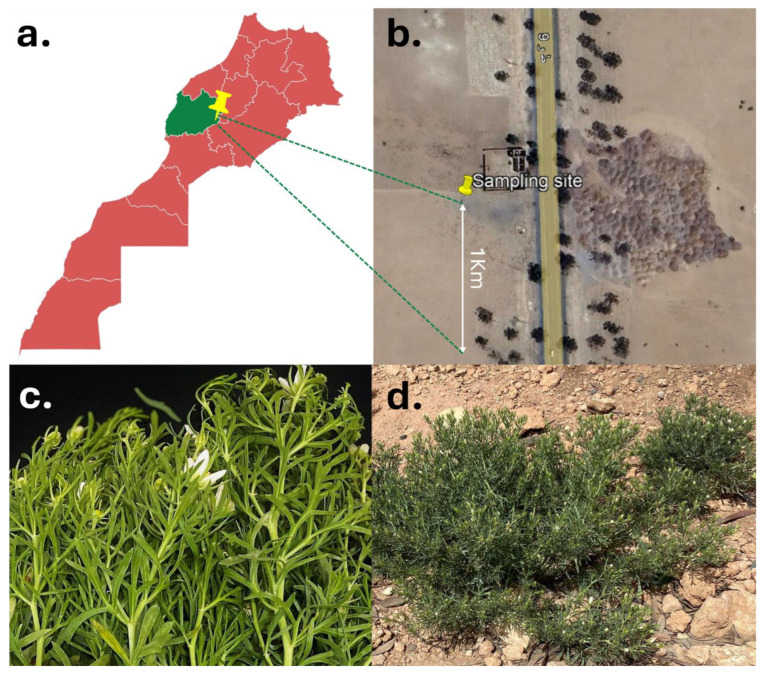
Geographical location of the host plant at the arid sampling site. (**a**) Map of Morocco showing the sampling region (green) and the precise collection locality (yellow pin). (**b**) Satellite view of the sampling site in Nzalat Laadam, illustrating the 1 km transect used for specimen collection (Google Earth). (**c**) Healthy leaves of naturally occurring *Peganum harmala*. (**d**) High-resolution photograph of natural *P. harmala* populations in their native arid habitat.

**Figure 2 microorganisms-13-02843-f002:**
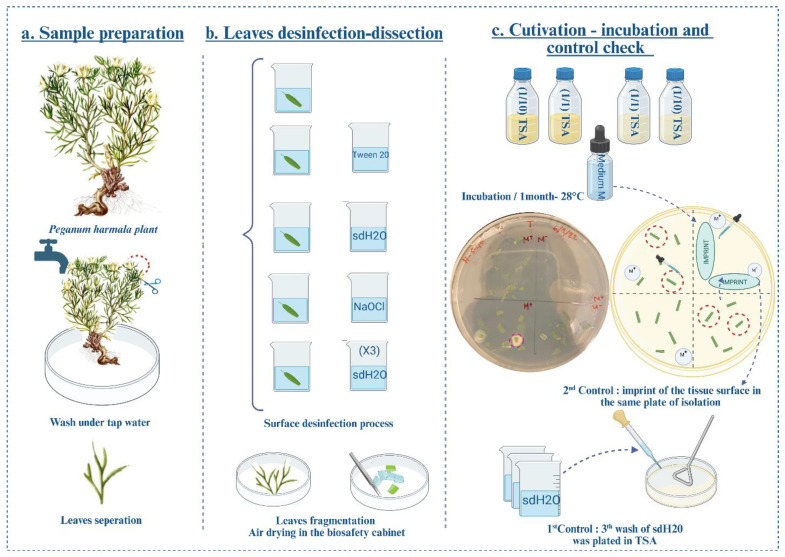
Graphical workflow of the bacterial endophyte isolation procedure. (**a**) Sample pre-processing. (**b**) Surface disinfection and tissue fragmentation. (**c**) Cultivation, contamination control, and incubation. M+: TSA enriched with M medium; M−: TSA without M-medium enrichment. Dashed circles in Petri dishes indicate microbial growth in close proximity to the foliar tissue.

**Figure 3 microorganisms-13-02843-f003:**
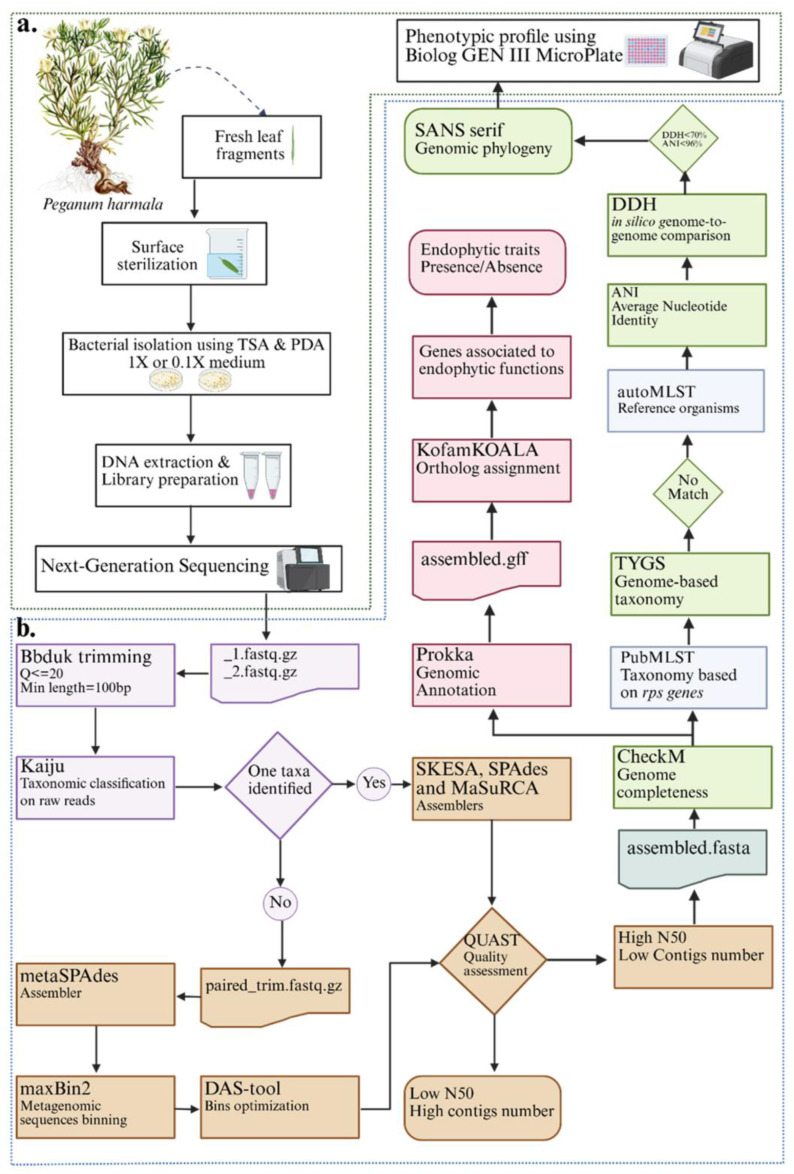
Experimental workflow and bioinformatic pipelines for whole-genome sequencing of endophytic bacteria from *P. harmala* leaves. (**a**) Isolation, DNA extraction, and sequencing steps. (**b**) Bioinformatics workflow and phenotypic profiling pipeline. Purple background: pre-processing; brown background: assembly and quality assessment; blue background: molecular typing characterization; green background: genomic taxonomy and species delineation; red background: functional annotation and trait mining.

**Figure 4 microorganisms-13-02843-f004:**
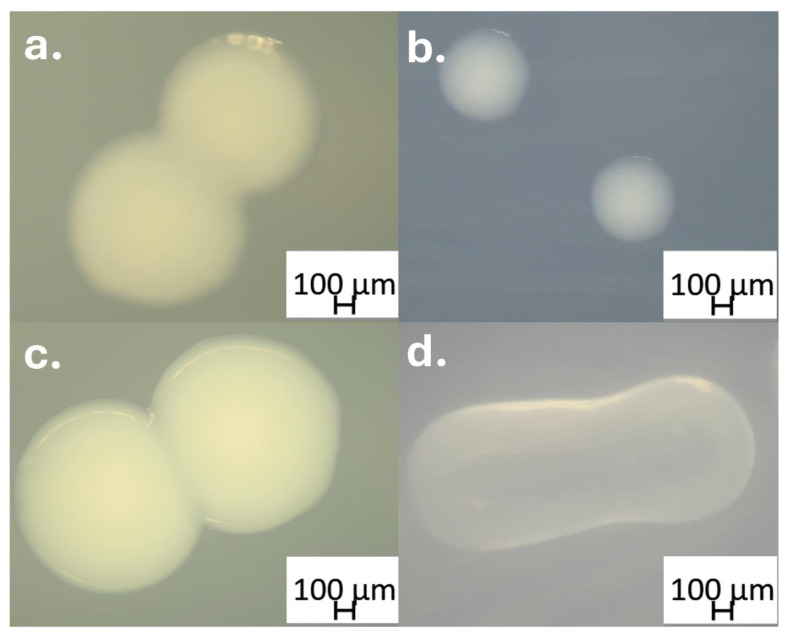
Colony macro-morphology of the two *Acinetobacter* isolates. Panels (**a**,**b**) show the colony morphology of *Acinetobacter endophylla* sp. nov. strain AGC35 grown on TSA (1×) and TSA (1/10), respectively. Panels (**c**,**d**) show the colony morphology of *Acinetobacter pittii* strain AGC59 on TSA (1×) and TSA (1/10), imaged using a microscope at 4× magnification.

**Figure 5 microorganisms-13-02843-f005:**
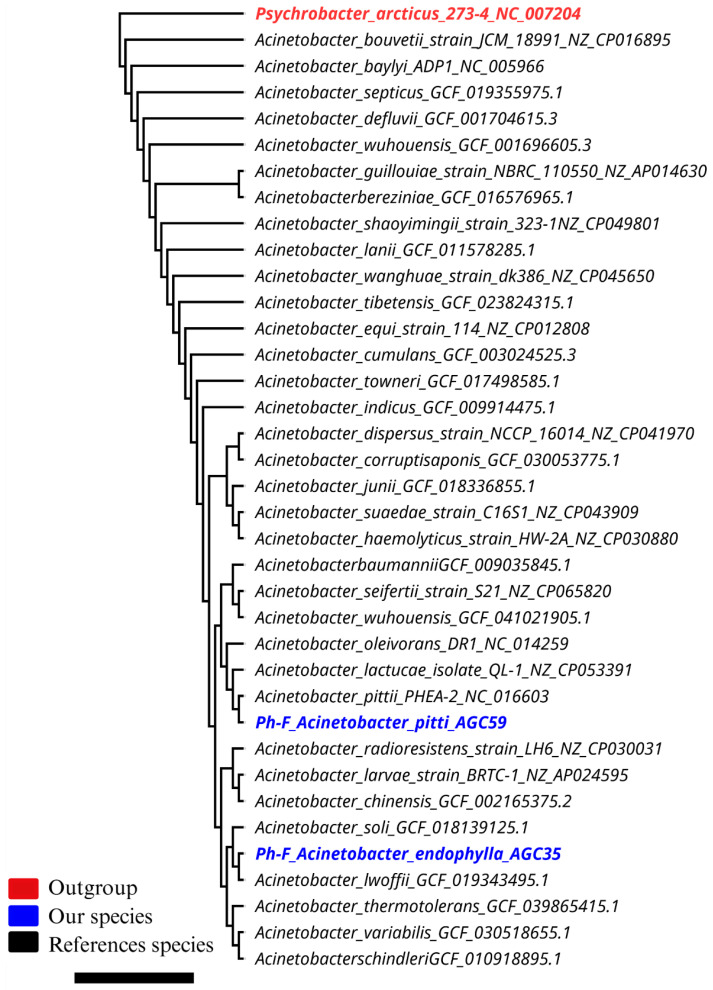
Rooted phylogenomic tree of the genus *Acinetobacter* (36 complete genomes). Maximum-likelihood analysis positions isolate AGC35 (candidate *Acinetobacter endophylla* sp. nov.) as a distinct lineage closest to the *A. lwoffii* clade, and AGC59 within the *A. pittii* cluster.

**Figure 6 microorganisms-13-02843-f006:**
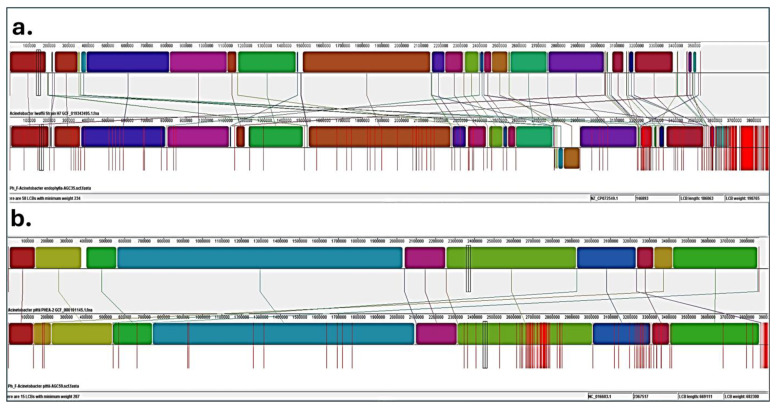
Comparative genomic synteny of the two *Acinetobacter* isolates. (**a**) Extensive genome rearrangement in AGC35 compared with *A. lwoffii* strain H7 (58 LCBs). (**b**) Conserved synteny in AGC59 relative to *A. pittii* PHEA-2 (15 LCBs). In the whole-genome alignment visualization, each colored block represents a Locally Collinear Block (LCB), corresponding to a homologous region conserved across the genomes. Blocks sharing the same color indicate positional homology. The vertical placement of a block (above or below the central axis of each track) indicates its orientation relative to the first genome (forward or inverted, respectively).

**Figure 7 microorganisms-13-02843-f007:**
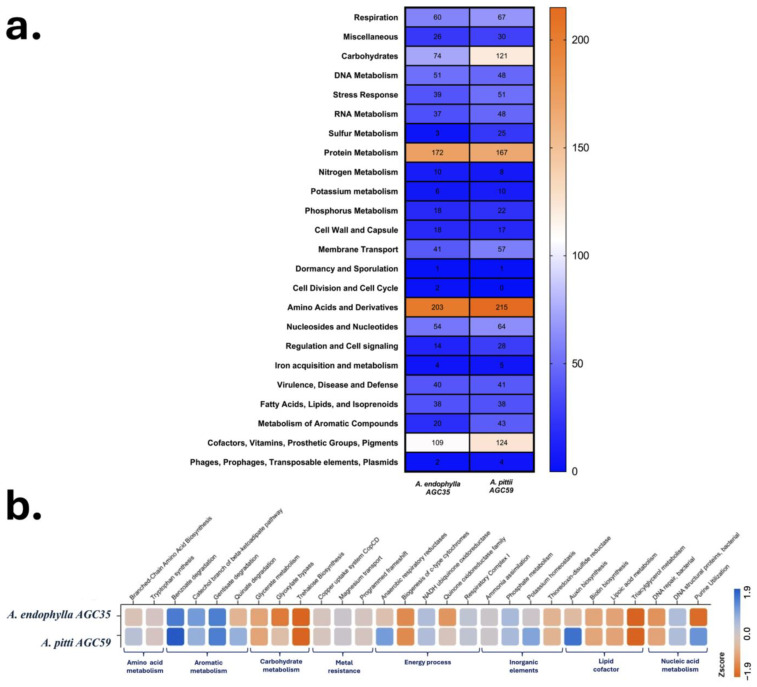
Functional and metabolic architecture of *A. endophylla* sp. nov. AGC35 and *A. pittii* AGC59. (**a**) Functional capabilities based on KEGG/KO gene counts grouped by major metabolic categories. (**b**) Z-score heatmap of key metabolic subsystems (amino acids, aromatic compounds, carbohydrates, energy, ion homeostasis, lipids/cofactors, nucleic acids).

**Figure 8 microorganisms-13-02843-f008:**
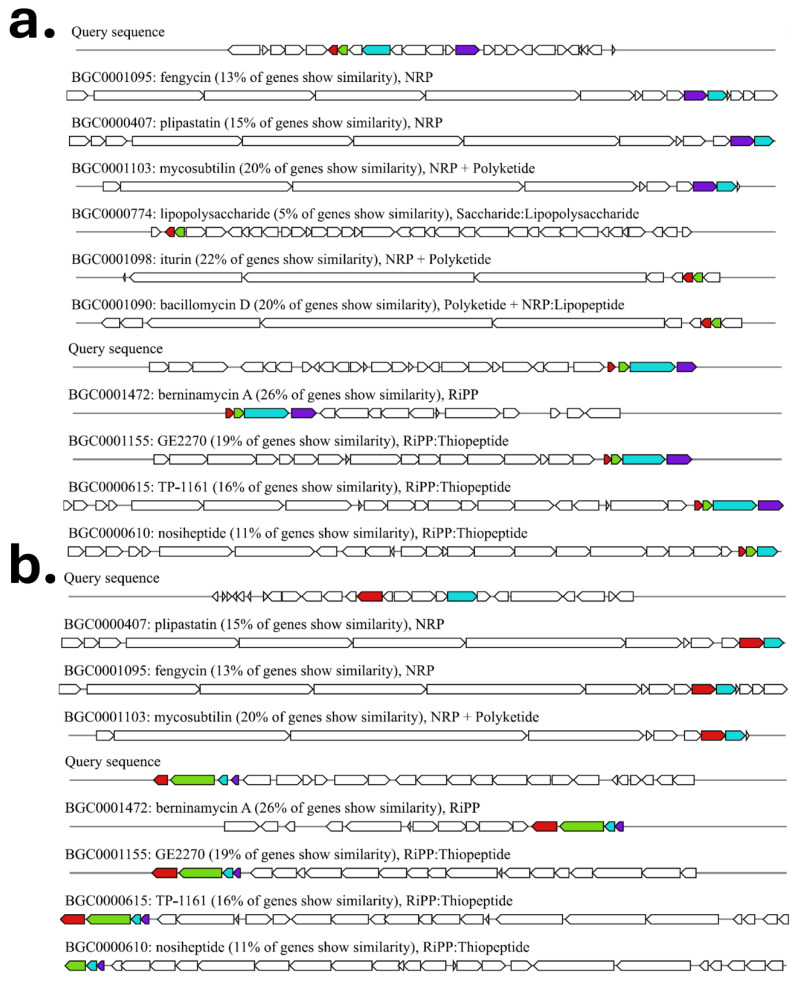
Biosynthetic gene cluster (BGC) profiles predicted using antiSMASH. (**a**) BGCs identified in *A. endophylla* AGC35, including NRPS/PKS and RiPP/thiopeptide clusters with similarity percentages to known BGCs. (**b**) BGCs identified in *A. pittii* AGC59.

**Figure 9 microorganisms-13-02843-f009:**
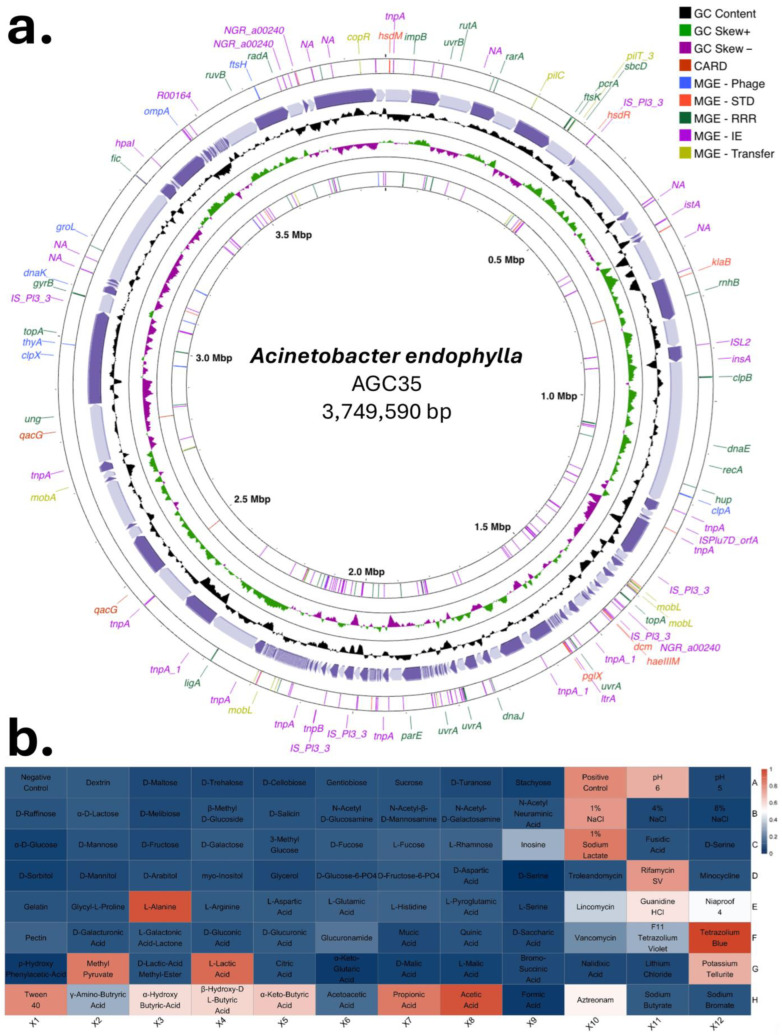
Circular genome map and phenotypic profiling of *A. endophylla* sp. nov. AGC35. (**a**) Circular genome visualization generated with Proksee, showing GC skew, genomic islands, and key functional clusters. (**b**) Biolog GEN III metabolic profile illustrating carbon-source utilization and chemical-resistance patterns (positive/negative reactions).

**Figure 10 microorganisms-13-02843-f010:**
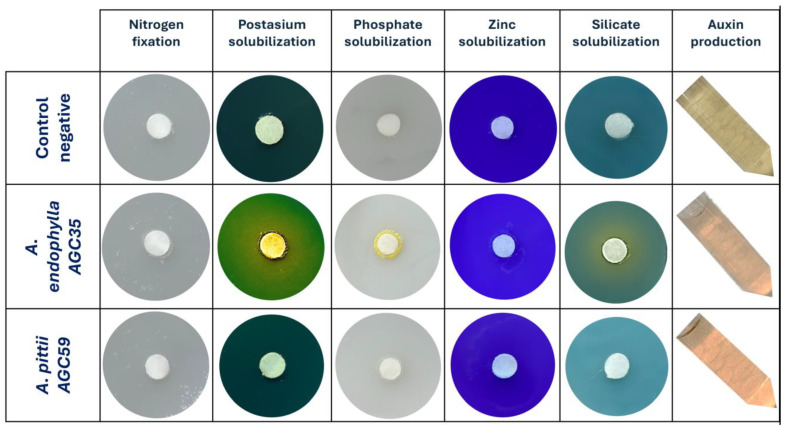
Plant-growth-promoting traits of the *Acinetobacter* isolates. Qualitative assays for nitrogen cycling, and the solubilization of potassium, phosphate, zinc, and silicate. Indole-3-acetic acid (IAA) production is shown per column. Rows represent negative control, AGC35, and AGC59. The strain’s ability to solubilize the inorganic element was confirmed by a virage color in the solid medium, indicated by the formation of a halo zone.

**Table 1 microorganisms-13-02843-t001:** Taxonomic classification of *Acinetobacter* isolates from *Peganum harmala.*

Bacterial Species (Strain)	Taxonomy Classification		Genome Comparisons		
	MLST Analysis	Closest Relative	MLST Analysis	Closest Relative	MLST Analysis
*Acinetobacter endophylla* sp. nov. (AGC35)	92%	*Acinetobacter lwoffii*	92%	*Acinetobacter lwoffii*	92%
*Acinetobacter pitti* (AGC59)	100%	*Acinetobacter pitti*	100%	*Acinetobacter pitti*	100%

## Data Availability

Accession numbers for the 16S rRNA gene sequences are available in GenBank under PV739381 and PV739392. Genome sequence data for the bacterial isolates have been deposited in the Sequence Read Archive (SRA) under SRR29855794, and SRR2985573. The type strain of the novel species has been deposited in the Moroccan Coordinated Collections of Micro-organisms (CCMM) (https://www.ccmm.ma, accessed on 15 November 2025), with accession numbers B1335^T^ (*Acinetobacter endophylla* sp. nov.). The genome assemblies are publicly accessible on Zenodo (https://zenodo.org/ accessed on 15 November 2025) under the following DOIs: https://doi.org/10.5281/zenodo.17615479.

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
