# Peer review of "Unveiling *Acinetobacter endophylla* sp. nov.: A Specialist Endophyte from *Peganum harmala* with Distinct Genomic and Metabolic Traits"

_microorganisms, 2025, doi:10.3390/microorganisms13122843_

Round 1

Reviewer 1 Report

Comments and Suggestions for Authors

The MS “Unveiling Acinetobacter endophylla sp. nov.: A Specialist Endo-phyte from Peganum harmala with Distinct Genomic and Meta-bolic Traits” report an important finding and is well written however, it needs a major overhaul before publication.

  1. Remove abbreviation from abstract and add future scope of work alongwith recommendations.
  2. In introduction in the last of paragraph, add how your work is novel than earlier work done on this topic.
  3. 1. Plant Habitat and Isolation of Endophytic Bacteria: which plant part you collected not clear, and what vegetative shoot or flower.
  4. Which method you used for the isolation of bacterial species did you repeat aanlysis of each test ?
  5. Write details about the DNA extraction, PCR amplification (PCR conditions and primes used in this study), PCR conditions etc.
  6. How many strains you used for PGP activity and why only three parameters tested not ACC deaminase, phosphate solubilization, ammonia production etc.
  7. Data analysis section is missing, mention about the repetition of experiment.
  8. Results are ok but mention real p value in isolation and PGP activity sections
  9. Remove subheadings from discussion and add future scope of this study in sustainable agriculture.
  10. The phylogenetic tree methodology should be elaborated and add bootstrap value in the tree and root. Also add type isolate.

Author Response

Comment 1: Remove abbreviation from abstract and add future scope of work along with recommendations.

Response  1: All abbreviations in the abstract (e.g., ANI, dDDH, PGP, PK, Si, AntiSMASH, BGCs) have been replaced with their full scientific terms for improved clarity. Additionally, we have added a concise statement outlining the future scope of the research, including the exploration of newly identified biosynthetic gene clusters and the evaluation of the isolates for applications in sustainable agriculture and bioactive compound development.

Comment 2: Plant Habitat and Isolation: which plant part you collected not clear, and what vegetative shoot or flower.

Response 2: We have clarified that aerial shoots of Peganum harmala were collected during the flowering stage. Bacterial strains were isolated from surface-sterilized healthy vegetative shoot segments, specifically leaf tissue.

Comment 3: Which method you used for the isolation of bacterial species? Did you repeat analysis of each test?

Response 3: We used a tissue-plating technique following surface sterilization of aerial shoot fragments, which were plated on TSA medium as described in the manuscript. The workflow is summarized in a schematic included in the revised version. Each Petri dish contained four quadrants: one imprint control and three replicates with sterilized tissue pieces.
All PGP assays were performed in quadruplicate for each strain to ensure reproducibility.

Comment 4: Write details about DNA extraction, PCR amplification, conditions, primers, etc.

Response 4: Additional details have been incorporated. Please refer to the corrections highlighted in red.

Comment 5: How many strains were used for PGP activity and why only three parameters tested? Not ACC deaminase, ammonia production, etc.

Response 5: PGP characterization was performed on the two target strains, Acinetobacter endophylla sp. nov. AGC35 and Acinetobacter pittii AGC59, since they were the focus of the genomic and functional analyses. Six major PGP traits were included: nitrogen fixation, phosphate solubilization, potassium solubilization, zinc solubilization, silicate solubilization, and auxin production. These represent core macro- and micronutrient mobilization markers suitable for initial screening. We agree that traits such as ACC deaminase and ammonia production are important, and we plan to incorporate these assays in future studies evaluating their biofertilizer and biocontrol potential.

Comment 6: Data analysis section is missing; mention repetition of experiment.

Response 6: We have added a data analysis subsection specifying that all PGP assays were conducted with four replicates per strain, and that interpretations were based on reproducible qualitative outcomes.

Comment 7: Mention real p-value in isolation and PGP activity sections.

Response 7: The isolation procedure was qualitative, based on presence/absence of bacterial g rowth following surface sterilization, and therefore does not produce quantitative data suitable for statistical analysis or p-value computation. This clarification has been added.

Comment 8: Remove subheadings from Discussion and add future scope in sustainable agriculture.

Response 8: Subheadings in the Discussion have been removed. A new paragraph has been added outlining the future scope, including deeper physiological and genomic characterization, assessment as PGP inoculants, exploration of biosynthetic gene clusters for novel metabolites, and potential integration into sustainable agricultural biotechnology.

Comment 9: Elaborate on phylogenetic tree methodology; add bootstrap values, root, and type isolate.

Response 9: The phylogenomic methodology has been expanded to describe the computation of ANI matrices (FastANI) and tree construction (FastME, minimum evolution).
We also clarified that bootstrap support values cannot be generated for ANI-based trees because ANI produces aggregated genome-wide similarity values without position-specific alignments required for resampling. In the revised version, the tree has been properly rooted and includes relevant type isolates to strengthen taxonomic interpretation.

Reviewer 2 Report

Comments and Suggestions for Authors

thank you for a well edited paper-  a few problems are noted by sticky notes

please also see supp table  3 and 4  spellings  length  absence presence

interesting how similar yet dissimilar the two isolate are

could you add whether these two isolates are compatible for their growths? or is one antagonistic? 

Author Response

General comment: Thank you for a well-edited paper. A few problems are noted by sticky notes. Please also see supplementary tables 3 and 4 for spelling, length, absence/presence.

Response: We thank the reviewer for their careful examination. All corrections indicated in the sticky notes and in Supplementary Tables 3 and 4 have been addressed, including spelling, formatting, and consistency in presence/absence notation.

Comment 1: Do not understand: you state limited CHO utilization but then state lives on host-derived simple sugars and organic acids.

Response 1: We agree that the initial phrasing may have appeared contradictory, and we appreciate the opportunity to clarify. When we state that A. endophylla exhibits limited carbon utilization, we are referring specifically to its restricted capacity to metabolize a broad range of complex carbon sources in standardized biochemical assays. This limitation is, however, fully aligned with ecological specialization toward a narrow set of readily available substrates, particularly simple sugars and organic acids that are naturally abundant in foliar tissues.

Endophytic bacteria commonly display reduced metabolic breadth while showing enhanced efficiency for specific host-derived substrates. This reflects adaptation to a nutrient environment dominated by low-molecular-weight compounds rather than heterogeneous external carbon sources.

To improve clarity, we revised the paragraph as follows:

“From genome to phenotype, functional validation consistently supports the ecological specialization of A. endophylla. Phenotypic profiling reveals a narrow carbon-utilization spectrum on standardized biochemical panels, indicating limited capacity to metabolize diverse or complex substrates. This restricted metabolic breadth aligns with adaptation to the endo-foliar niche, where bacteria predominantly encounter simple host-derived compounds such as monosaccharides, organic acids, and amino acids rather than heterogeneous environmental carbon sources. In addition to its limited carbon utilization, A. endophylla displays sensitivity to osmotic stress and tolerance to specific ionic stresses (sodium butyrate, sodium bromate, Niaproof 4), further reflecting further reflecting selective pressures characteristic of the foliar environment, where microbes must rely on host-derived nutrients and withstand biotic stresses, as reported by Mujumdar, et al. [62].”

Minor note: Please also see Supplementary Tables 3 and 4 — spellings, lengths, absence/presence: I didn’t know how to adjust it.

Response: We corrected spelling inconsistencies, standardized the terminology (e.g., “presence/absence”), and ensured consistent formatting for gene lengths and annotations.Supplementary Tables 3 and 4.

Comment: Interesting how similar yet dissimilar the two isolates are. Could you add whether these two isolates are compatible in their growth, or is one antagonistic?

Response: The two isolates are compatible, as evidenced by the dual-culture assays shown in the recto-verso images. No antagonistic activity was observed under the tested conditions. We have added a brief statement in the Results section to clarify this point.

Comment: are these two strains compatible on dual plate culture?

Response: Yes, a compatibility test performed on TSA medium with both strains showed no growth inhibition at their intersection, indicating that the two isolates are compatible (Fig. S2). We added this information the conclusion.

Round 2

Reviewer 1 Report

Comments and Suggestions for Authors

authors addressed my all comments, the MS can be accepted.